# Analysis of Volatile Constituents in *Platostoma palustre* (Blume) Using Headspace Solid-Phase Microextraction and Simultaneous Distillation-Extraction

**DOI:** 10.3390/foods8090415

**Published:** 2019-09-14

**Authors:** Tsai-Li Kung, Yi-Ju Chen, Louis Kuoping Chao, Chin-Sheng Wu, Li-Yun Lin, Hsin-Chun Chen

**Affiliations:** 1Taoyuan District Agricultural Research and Extension Station, Council of Agriculture, Executive Yuan, Taoyuan 327, Taiwan; tlkung@tydais.gov.tw; 2Department of Cosmeceutics, China Medical University, Taichung 404, Taiwan; yc9429@hotmail.com (Y.-J.C.); kuoping@mail.cmu.edu.tw (L.K.C.); cswu@mail.cmu.edu.tw (C.-S.W.); 3Department of Food Science and Technology, Hungkuang University, Taichung 433, Taiwan

**Keywords:** Hsian-tsao, *Platostoma palustre* (Blume), headspace solid-phase microextraction (SPME), volatile components, simultaneous distillation-extraction (SDE)

## Abstract

Hsian-tsao (*Platostoma palustre* Blume) is a traditional Taiwanese food. It is admired by many consumers, especially in summer, because of its aroma and taste. This study reports the analysis of the volatile components present in eight varieties of Hsian-tsao using headspace solid-phase microextraction (HS-SPME) and simultaneous distillation-extraction (SDE) coupled with gas chromatography (GC) and gas chromatography-mass spectrometry (GC/MS). HS-SPME is a non-heating method, and the results show relatively true values of the samples during flavor isolation. However, it is a kind of headspace analysis that has the disadvantage of a lower detection ability to relatively higher molecular weight compounds; also, the data are not quantitative, but instead are used for comparison. The SDE method uses distillation 2 h for flavor isolation; therefore, it quantitatively identifies more volatile compounds in the samples while the samples withstand heating. Both methods were used in this study to investigate information about the samples. The results showed that Nongshi No. 1 had the highest total quantity of volatile components using HS-SPME, whereas SDE indicated that Taoyuan Mesona 1301 (TYM1301) had the highest volatile concentration. Using the two extraction methods, 120 volatile components were identified. Fifty-six volatile components were identified using HS-SPME, and the main volatile compounds were α-pinene, β-pinene, and limonene. A total of 108 volatile components were identified using SDE, and the main volatile compounds were α-bisabolol, β-caryophyllene, and caryophyllene oxide. Compared with SDE, HS-SPME sampling extracted a significantly higher amount of monoterpenes and had a poorer detection of less volatile compounds, such as sesquiterpenes, terpene alcohols, and terpene oxide.

## 1. Introduction

Hsian-tsao (*Platostoma palustre* Blume, also known as *Mesona procumbens* Hemsl. [1]), also called Liangfen Cao or black cincau, belongs to the family Lamiaceae. It is an annual plant that is mainly distributed in tropical and subtropical regions, including Taiwan, southern China, Indonesia, Vietnam, and Burma [2]. Hsian-tsao tea, herbal jelly, and sweet soup with herbal jelly are popular during the summer, and heated herbal jelly is admired by many Taiwanese, especially in winter, because of its aroma and taste. In Indonesia, janggelan (*Mesona palustris* BL) has also been made into a herbal drink and a jelly-type dessert [3]. Hsian-tsao is also used as a remedy herb in folk medicine and is supposed to be effective in treating heat-shock, hypertension, diabetes, liver diseases, and muscle and joint pains [4,5].

Hsian-tsao contains polysaccharides (gum) with a unique aroma and texture. Most research has investigated the gum of Hsian-tsao [2,6,7,8]; however, there are only a few studies of Hsian-tsao aroma. Wei et al. [9] identified 59 volatile compounds in *Mesona* Benth extracted using solvent extraction. They also reported that the important constituents were caryophyllene oxide, α-caryophyllene, eugenol, benzene acetaldehyde, and 2,3-butanedione. Deng et al. [10] reported the chemical constituents of essential oil from *Mesona chinensis* Benth (also known as *P. palustris* Blume [1]) using GC/MS. The major constituents were *n*-hexadecanoic acid, linoleic acid, and linolenic acid. Lu et al. [11] analyzed the volatile oil from *Mesona chinensis* Benth using GC/MS. The results indicated the main components were chavibetol, *n*-hexadecanoic acid, and α-cadinol.

Simultaneous distillation-extraction (SDE) is a traditional extraction method that was introduced by Likens and Nickerson in 1946. SDE combines the advantages of liquid–liquid extraction and steam distillation methods. It is widely used for the extraction of essential oils and volatile compounds [12,13]. In the flavor field, this technique is recognized as a superior extraction method compared to other methods, such as solvent extraction or distillation. Moreover, Gu et al. [14] indicated that SDE has excellent reproducibility and high efficiency compared with traditional extraction methods.

Headspace solid-phase microextraction (HS-SPME) is a non-destructive and non-invasive method that avoids artifact formation and solvent impurity contamination [15]. HS-SPME is a fast, simple, and solventless technique [16,17,18]. HS-SPME can integrate sampling, extraction, concentration and sample introduction into a single uninterrupted process, resulting in high sample throughput [19].

This study aimed to identify the volatile constituents in different varieties of Hsian-tsao and the differences in extraction methods (HS-SPME and SDE). The differences in volatile compounds caused by heating are discussed. The results from this study provide a reference for the food, horticultural, and flavor industries.

## 2. Materials and Methods 

### 2.1. Plant Materials

A total of eight varieties of Hsian-tsao from throughout Taiwan were used in this study (Table 1): Nongshi No. 1 from Tongluo Township in Miaoli County; Taoyuan No. 2 from Shoufeng Township in Hualien County; Chiayi strain from Shuishang Township in Chiayi County; Taoyuan No. 1, TYM1301, and TYM1302 from Guanxi Township in Hsinchu County; and TYM1303 and TYM1304 from Shuangxi District in New Taipei City. Eight varieties of Hsian-tsao were grown at the Sinpu Branch Station (Sinpu Township in Hsinchu County) of Taoyuan District Agricultural Research and Extension Station. The identities of the plants were confirmed by Tsai-Li Kung (Chief of the Sinpu Branch Station). After shade drying, dried samples were stored at room temperature for one year before the experiment was conducted.

### 2.2. Methods

#### 2.2.1. Optimization of the HS-SPME Procedure

The method used was modified from those of Yeh et al. [20]:Comparisons of SPME fiber coatings: five different coated SPME fibers, 85 μm polyacrylate (PA), 100 μm polydimethylsiloxane (PDMS), 65 μm polydimethylsiloxane/divinylbenzene (PDMS/DVB), 75 μm carboxen/polydimethylsiloxane (CAR/PDMS), and 50/30 μm divinylbenzene/carboxen/polydimethylsiloxane (DVB/CAR/PDMS), (Supelco, Inc., Bellefonte, PA, USA) were used for the aroma extraction. Samples (Nongshi No. 1) were placed into a homogenizer (WAR7012S 7-Speed Blender 1 Qt. 120 V) (Waring commercial, Torrington, CT, USA). After being homogenized for 30 s, 1 g of homogenized samples was put into a 7 mL vial (Hole Cap Polytetrafluoroethene/Silicone Septa) (Supelco, Inc., Bellefonte, PA, USA) and sealed. The SPME method was used to extract the aroma components. The extraction temperature was 25 ± 2 °C and the extraction time was 40 min. This experiment and all other experiments in this study were performed in triplicates.Comparisons of the extraction times: The above-mentioned optimal extraction fiber was used in the comparison of the extraction times. The tested extraction times were 10 min, 20 min, 30 min, 40 min, or 50 min, and the extraction temperature was maintained at 25 ± 2 °C. Sample preparation steps were the same as above.

#### 2.2.2. Analysis of the Volatile Compounds

Analysis of the volatile compounds using HS-SPME extraction: a 50/30 μm divinylbenzene/carboxen/polydimethylsiloxane (DVB/CAR/PDMS) fiber (Supelco, Inc., Bellefonte, PA, USA) was used for aroma extraction. The eight different Hsian-tsao varieties were used as samples. Each sample was homogenized as described above in Section 2.2.1 (1 g was placed in a 7 mL vial (hole cap PTFE/silicone septa)). The SPME fiber was exposed to each sample for 40 min at 25 ± 2 °C; then, each sample was injected into a gas chromatograph injection unit. The injector temperature was maintained at 250 °C and the fiber was held for 10 min. The peak area of a volatile compound or total volatile compounds from the integrator was used to calculate the relative contents.Analysis of volatile compounds by SDE extraction: 100 g samples of Hsian-tsao were cut with scissors into pieces approximately 1–3 cm in size and were then homogenized for 2 min with 2 L of deionized water and were placed into a 5-L round-bottom flask. The flask was attached to a simultaneous distillation-extraction apparatus and 100 °C steam was used as the heat source and passed through the sample. A 50 mL volume of solvent was prepared by mixing *n*-pentane/diethyl ether (1:1, *v*/*v*) into a pear-shaped flask, then placing it in a 40–45 °C water bath. This distillation circulation continued for 2 h, and the collected solvent extract was added to 200 μL of an internal standard solution of cyclohexyl acetate, and an internal standard was used to obtain the weight concentration of volatile compound in the sample; also, anhydrous sodium sulfate was used to remove the water. Lastly, the distillation column (40 °C, 1 h, 100 cm glass column) was used to volatilize the solvent and the condensed volatile component extracts were collected.GC analysis of the volatile compounds was conducted using a 7890A GC (Agilent Technologies, Palo Alto, CA, USA) equipped with a DB-1 (60 m × 0 .25 mm i.d. × 0.25 μm film thickness, Agilent Technologies) capillary column and a flame ionization detector. The injector and detector temperatures were maintained at 250 °C and 300 °C, respectively. The oven temperature was held at 40 °C for 1 min and then raised to 150 °C at 5 °C/min and held for 1 min, and then increased from 150 to 200 °C at 10 °C/min and held for 11 min. The carrier gas (nitrogen) flow rate was 1 mL/min. The Kovats indices were calculated for the separated components relative to a C_5_–C_25_
*n*-alkanes mixture [21]. The purpose gas chromatography-flame ionization detector (GC-FID) was used both for retention indices (RI) comparison and quantitation of peak areas.GC-MS analysis of volatile compounds were identified using an Agilent 7890B GC equipped with DB-1 (60 m × 0.25 mm i.d. × 0.25 μm film thickness) fused silica capillary column coupled to an Agilent model 5977 N MSD mass spectrometer (MS). The GC conditions in the GC-MS analysis were the same as in the GC analysis. The carrier gas (helium) flow rate was 1 mL/min. The electron energy was 70 eV at 230 °C. The constituents were identified by matching their spectra with those recorded in an MS library (Wiley 7N, John Wiley & Sons, Inc. New Jersey, NJ, USA). In addition, the constituents were confirmed by comparing the Kovats indices or GC retention time data with those of authentic standards or data published in the literature. The GC and GC-MS methods used were modified from those of Yeh et al. [20].Statistical Analysis: Each sample was extracted in triplicate and the concentration of volatile compounds was determined as the mean value of three repetitions. The data were subjected to a monofactorial variance analysis with Duncan’s multiple range method with a level of significance of *p* < 0.05 (SPSS Base 12.0, SPSS Inc., Chicago, IL, USA).

## 3. Results

### 3.1. Comparisons of the Optimization Conditions of HS-SPME

#### 3.1.1. SPME Fiber Selection

The performance of five commercially available SPME fibres: 50/30-μm DVB/CAR/PDMS, 65-μm PDMS/DVB, 75-μm CAR/PDMS, 100-μm PDMS, and 85-μm PA were used to extract the volatile components of Nongshi No. 1. The 50/30-μm DVB/CAR/PDMS fiber extracted more total volatile components than the other fibers (Figure 1).

Ducki et al. [22] evaluated four different types of SPME fibers (100-μm PDMS, 65-μm PDMS/DVB, 75-μm CAR/PDMS, and 50/30-μm DVB/CAR/PDMS) for the headspace analysis of volatile compounds in cocoa products. The SPME fiber coated with 50/30-μm DVB/CAR/PDMS afforded the highest extraction efficiency. Silva et al. [23] compared the performance of six fibers (PDMS, PDMS/DVB, CW/DVB, PA, CAR/PDMS, and DVB/CAR/PDMS) and found that DVB/CAR/PDMS was the most effective SPME fiber for isolating the volatile metabolites from *Mentha* × *piperita* L. fresh leaves based on the total peak areas, reproducibility, and number of extracted metabolites. Yeh et al. [20] reported the volatile components in *Phalaenopsis* Nobby’s Pacific Sunset, and the optimal extraction conditions were obtained using a DVB/CAR/PDMS fiber.

The 50/30-μm DVB/CAR/PDMS was revealed to be the most suitable and was subsequently used in all further experiments.

#### 3.1.2. HS-SPME Extraction Time

The optimal SPME fiber (50/30-μm DVB/CAR/PDMS) was used to extract Nongshi No. 1 at 25 ± 2 °C, and the extraction times from 10 to 50 min were investigated. The total peak area increased from 10–40 min and reached the peak at 40 min (Figure 2). Silva and Câmara [23] promoted the higher extraction efficiency, corresponding to the higher GC peak areas and the number of identified metabolites. This higher extraction efficiency was achieved using: DVB/CAR/PDMS coating fiber, and 40 °C and 60 min as the extraction temperature and extraction time, respectively. Zhang et al. [24] also obtained optimum extraction conditions, which were using 50/30-μm DVB/CAR/PDMS fiber for 40 min at 90 °C. According to the obtained results, 40 min was selected as the optimal extraction time. 

### 3.2. Analyses of the Volatiles of Eight Varieties of Hsian-Tsao Using HS-SPME 

As shown in Figure 3, the total peak areas of volatile components was the highest in Nongshi No. 1 and lowest in TYM1304. Volatile compounds in eight varieties of Hsian-tsao were analyzed using headspace solid-phase microextraction (HS-SPME), which was coupled with GC and GC/MS. Table 2 shows a total of 56 compounds that were identified. Monoterpene compounds were the most abundant compounds in the Hsian-tsao analyzed using HS-SPME/GC. The main volatile components of Nongshi No. 1, Chiayi strain, TYM1302, and TYM1303 were β-pinene (43–50%), α-pinene (10–24%), and limonene (4–9%). β-Pinene (36–42%), α-pinene (15–17%), and β-caryophyllene (11%) were the main components from Taoyuan No. 2 and TYM1301. The main components of Taoyuan No. 1 were β-pinene (23%), limonene (21%), and α-pinene (11%). Limonene (32%), β-caryophyllene (13%), and sabinene (7%) were the main components from TYM1304. TYM1304 contained the highest content of limonene (32%), followed by Taoyuan No. 1 (21%). Limonene is a citrus note, having a pungent green and lemon-like aroma [25,26]. The peak areas of α-pinene and β-pinene were the highest in TYM1302 (24% and 50%), whereas TYM1304 was lower than the other varieties. α-Pinene was described as having a fruity, piney, and turpentine-like aroma [27,28], and β-pinene was described as having a dry-woody, pine-like, and citrus aroma [29,30]. β-Caryophyllene was described as having a dry-woody, pine-like, and spicy aroma, and TYM1304 contained the highest content (13%), followed by TYM1301, and Taoyuan No. 2 (11%).

The eight varieties of Hsian-tsao shared 15 volatile components; the differences in percentage were: 1-octen-3-ol (trace–3%), hexanal (trace–4%), 1-octen-3-one (trace–1%), α-thujene (trace–3%), α-pinene (5–24%), sabinene (2–7%), β-pinene (2–50%), β-myrcene (trace–2%), α-terpinene (trace–2%), limonene (4–32%), α-ylangene (trace), α-copaene (trace–2%), β-elemene (1–4%), β-caryophyllene (3–13%), and α-caryophyllene (1–2%). Among them, hexanal was described as having a green and cut-grass aroma [31], and was responsible for green, apple, and green fruit perceptions [32]. Nongshi No. 1 contained the highest content of hexanal (4%).

### 3.3. Analysis of the Volatiles of Eight Varieties (Clones) of Hsian-Tsao Using SDE

As shown in Table 3, the volatile components content peaked in TYM1301 and was the lowest in TYM1303. Table 4 shows the results of the SDE analysis: 108 components were identified, including 11 aliphatic alcohols, 14 aliphatic aldehydes, 9 aliphatic ketones, 1 aliphatic ester, 3 aromatic alcohols, 2 aromatic aldehydes, 1 aromatic ketones, 2 aromatic esters, 4 aromatic hydrocarbons, 8 terpene alcohols, 2 terpene aldehydes, 2 terpene ketones, 10 monoterpenes, 22 sesquiterpenes, 1 terpene oxide, 7 hydrocarbons, 3 straight-chain acids, 3 furans, 2 methoxy-phenolic compounds, and 1 nitrogen-containing compound. Sesquiterpene compounds, terpene alcohols, and terpene oxide were the main compounds in the Hsian-tsao analyzed using SDE. The major components of Hsain-tsao (Nongshi No. 1, Taoyuan No. 1, and TYM1301) were α-bisabolol (59–144 mg/kg), caryophyllene oxide (9–28 mg/kg), and β-caryophyllene (21–54 mg/kg). β-Caryophyllene (21–56 mg/kg) and caryophyllene oxide (32–50 mg/kg) were the main compounds of Taoyuan No. 2 and TYM1302. α-Bisabolol (116 mg/kg), β-bisabolene (24 mg/kg), β-selinene (23 mg/kg), and β-caryophyllene (19 mg/kg) were the main components of the Chiayi strain. α-Bisabolol (144 mg/kg), α-bisabolol (43 mg/kg), and caryophyllene oxide (15 mg/kg) were the main components of TYM1303. β-Caryophyllene (53 mg/kg), β-elemene (35 mg/kg), α-selinene (15 mg/kg), and α-caryophyllene (13 mg/kg) were the main compounds in TYM1304. Nongshi No. 1, Taoyuan No. 1, Chiayi strain, TYM1301, and TYM1303 contained a higher content of α-bisabolol; Taoyuan No. 2 and TYM1304 contained a higher content of β-caryophyllene; and TYM1302 contained a higher content of caryophyllene oxide. Wei et al. [9] analyzed the volatile components of *Mesona* Benth, where they reported that the main components were caryophyllene oxide and caryophyllene, similar with these experimental results. β-Caryophyllene had a woody aroma, and Taoyuan No. 2 had the highest concentration, followed by TYM1301 and TYM1304 (53–56 mg/kg). 

### 3.4. Comparisons of the Differences between HS-SPME and SDE

Similar to Table 2, Table 4 shows the eight different Hsian-tsao varieties, along with the 120 components identified using HS-SPME and SDE, of which, 44 were found using both extraction methods, 12 (mainly α-terpinene, δ-3-carene, and *cis*-α-bergamotene) were identified using HS-SPME but not detected using SDE, and 64 (mainly nonanal, 6-methyl-3,5-heptadien-2-one, and gossonorol) were identified using SDE but not detected using HS-SPME. 

Table 5 and Figure 4 show that the monoterpene relative content was higher than that of sesquiterpene. Table 6 and Figure 5 show that the SDE samples had a high content of sesquiterpenes, terpene oxide, and terpene alcohols, but a lower content of monoterpenes than the SPME samples. Tersanisni and Berry [33] reported that certain hydrocarbon compounds, such as linalool and α-terpineol, as well as their hydrocarbon interactions, can be interrupted by heat stress, resulting in the induction of volatilization. We detected α-terpineol using SDE but by using HS-SPME. However, both methods identified terpene hydrocarbons as the major components. HS-SPME extracted more terpene hydrocarbons, and the majority was highly volatile monoterpenes with a low molecular weight. SDE extracted mainly sesquiterpenes with higher molecular weights. SDE also identified components that HS-SPME was unable to identify, such as straight-chain acids, aromatic ketones, aromatic esters, terpene aldehydes, terpene ketones, methoxy phenols, and nitrogen-containing compounds. Montserrat et al. [34] analyzed the volatile composition of white salsify (*Tragopogon porrifolius* L.) and found that SDE used high temperature and a long extraction time, and large quantities of volatile components were lost during the extraction process. Therefore, the SDE method may increase the low volatile compounds with a high molecular weight, such as sesquiterpenes and straight-chain acids. HS-SPME used shorter extraction times, so it was able to extract highly volatile monoterpenes with lower molecular weights. As such, HS-SPME is more appropriate for quality control. This study found that although HS-SPME was more rapid and SDE had a higher temperature and longer extraction time, SDE was able to extract more Hsian-tsao compounds; therefore, both methods can be used to complement each other. Yang et al. [35] compared HS-SPME with traditional methods in the analysis of *Melia azedarach* and reported that the HS-SPME method is a powerful analytic tool and is complementary to traditional methods for the determination of the volatile compounds in herbs. Comparing both techniques, HS-SPME samples were smaller (1 g) and did not require heating, the data was accurate, and involved less chemical reactions and changes, but the yield of larger molecules were lower, and the identified components were fewer, while SDE needed the use of 100 g of plant material and heating (2 h). The popularity of this method comes from the fact that volatiles with medium to high boiling points are recovered well. The aroma profile can be greatly altered via the formation of artifacts due to heating the sample during isolation. However, Hsian-tsao food needs to be processed using heat; therefore, by combining the HS-SPME and SDE methods of volatile compounds isolation, each isolation technique provides a part of the overall Hsian-tsao profile. 

## 4. Conclusions

This study determined the volatile components present in eight varieties of Hsian-tsao using HS-SPME and SDE methods. A total of 120 volatile components were identified, of which, 56 were verified using HS-SPME and 108 using SDE. HS-SPME extracted more monoterpenes; however, SDE extracted more sesquiterpenes and terpene alcohols, and a terpene oxide, such as β-caryophyllene, α-bisabolol, and caryophyllene oxide. SDE was able to detect more components, but HS-SPME analysis was more convenient. In the future, the two extraction methods can be used in a complementary manner for Hsian-tsao analysis and research.

## Figures and Tables

**Figure 1 foods-08-00415-f001:**
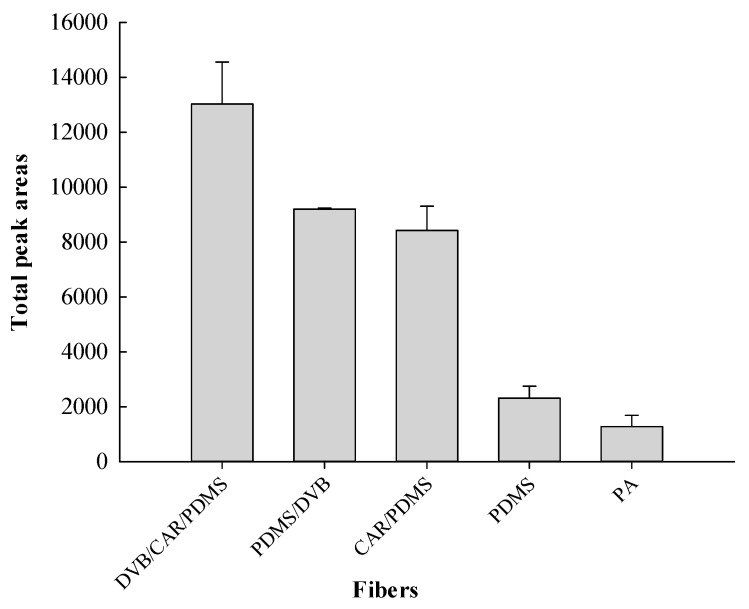
Comparisons of the total peak areas of total volatile compounds detected in the headspace of Nongshi No. 1 using different headspace solid-phase microextraction (HS-SPME) fibers.

**Figure 2 foods-08-00415-f002:**
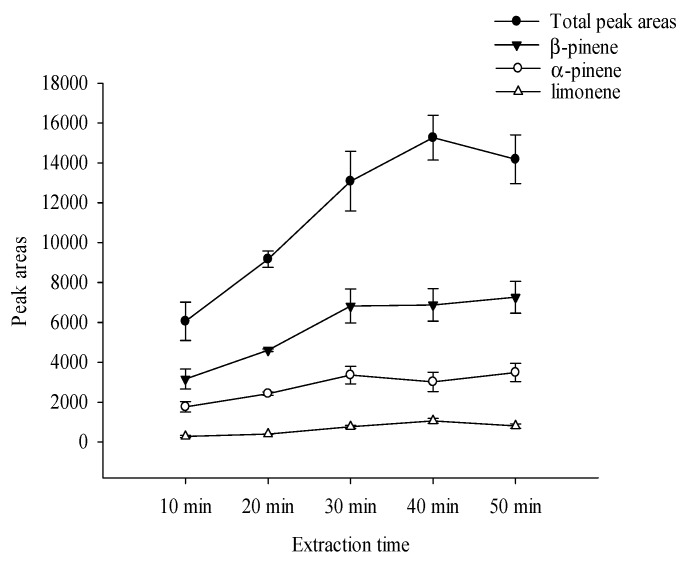
Comparisons of the peak areas of total volatile compounds and main components detected in the headspace of Nongshi No. 1 for different SPME extraction times at 25 °C using a DVB/CAR/PDMS fiber.

**Figure 3 foods-08-00415-f003:**
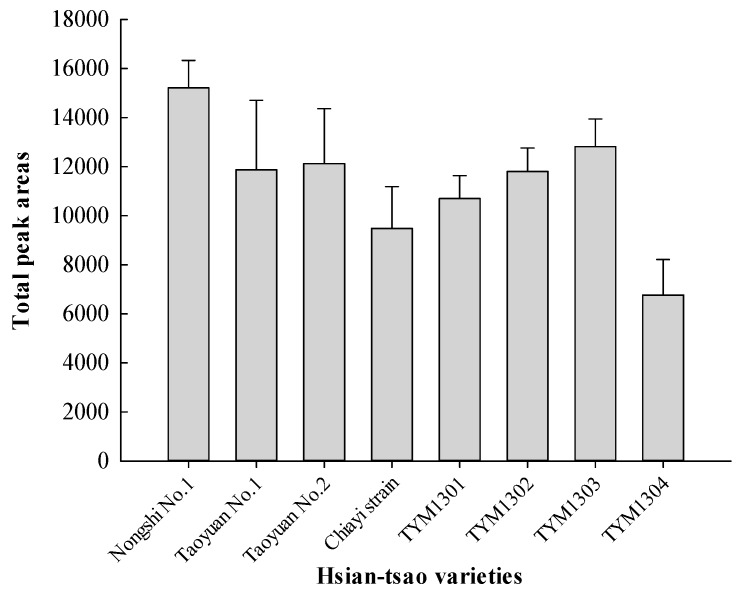
Comparisons of the total peak areas of total volatile compounds of eight varieties of Hsian-tsao (1 g) extracted at 25 °C for 40 min using a DVB/CAR/PDMS fiber. The peak area of a volatile compound or total volatile compounds from the integrator was used to calculate the relative contents using gas chromatography-flame ionization detector (GC-FID). The data corresponds to the mean ± standard deviation (SD) of triplicates.

**Figure 4 foods-08-00415-f004:**
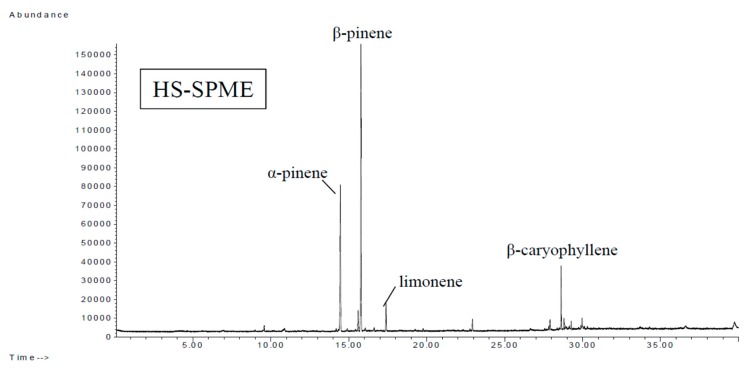
Total ion chromatogram of volatile components of Nongshi No. 1 determined using HS-SPME.

**Figure 5 foods-08-00415-f005:**
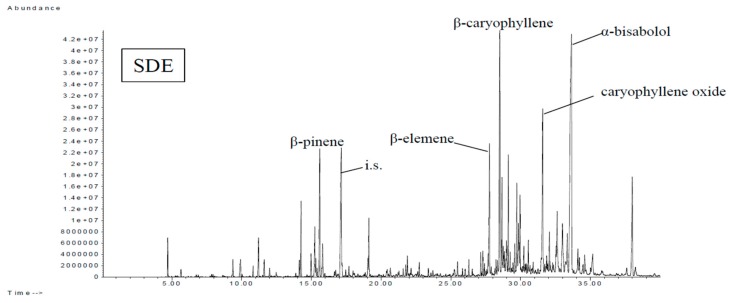
Total ion chromatogram of volatile components of Nongshi No. 1 determined using SDE.

**Table 1 foods-08-00415-t001:** The study of collections of taxa currently assigned to Hsian-tsao.

Varieties	Origin	Growing Locality
Nongshi No.1	Tongluo Township, Miaoli County	Sinpu Township, Hsinchu County
Taoyuan No.1	Guanxi Township, Hsinchu County	Sinpu Township, Hsinchu County
Taoyuan No.2	Shoufeng Township, Hualien County	Sinpu Township, Hsinchu County
Chiayi strain	Shuishang Township, Chiayi County	Sinpu Township, Hsinchu County
TYM1301	Guanxi Township, Hsinchu County	Sinpu Township, Hsinchu County
TYM1302	Guanxi Township, Hsinchu County	Sinpu Township, Hsinchu County
TYM1303	Shuangxi District, New Taipei City	Sinpu Township, Hsinchu County
TYM1304	Shuangxi District, New Taipei City	Sinpu Township, Hsinchu County

**Table 2 foods-08-00415-t002:** Comparisons of volatile compounds from eight varieties of Hsian-tsao extracted using HS-SPME.

Compound ^a^	RI ^b^	Relative Content (%) ^c^							
		Nongshi No. 1	Taoyuan No. 1	Taoyuan No. 2	Chiayi Strain	TYM1301	TYM1302	TYM1303	TYM1304
**Aliphatic alcohol**									
1-Octen-3-ol ^f^	962	0.29 ± 0.11	0.69 ± 0.24	1.16 ± 0.19	0.17 ± 0.03	2.25 ± 0.40	1.39 ± 0.28	1.61 ± 0.22	3.01 ± 0.82
**Aliphatic aldehydes**									
Hexanal ^f^	777	3.63 ± 0.52	1.85 ± 0.68	0.29 ± 0.10	0.39 ± 0.15	1.19 ± 0.26	1.15 ± 0.22	1.81 ± 0.24	1.74 ± 0.19
(*E*)-2-heptenal ^f^	931	tr ^d^	1.06 ± 0.20	tr	- ^e^	tr	tr	tr	tr
Decanal ^f^	1181	-	-	-	tr	-	-	-	-
**Aliphatic ketones**									
1-Octen-3-one ^f^	956	tr	0.14 ± 0.02	0.12 ± 0.02	tr	0.29 ± 0.03	tr	0.24 ± 0.02	0.47 ± 0.07
3-Octanone ^f^	965	-	-	-	tr	-	-	tr	tr
**Aliphatic ester**									
Linalyl isobutyrate	1387	0.32 ± 0.05	-	-	-	-	-	-	-
**Aromatic alcohol**									
Eugenol ^g^	1328	-	-	-	0.08 ± 0.02	-	-	-	-
**Aromatic aldehyde**									
Benzaldehyde ^f^	933	tr	-	-	tr	-	-	-	-
**Aromatic hydrocarbon**									
*p*-Cymene	1004	-	0.68 ± 0.05	-	tr	tr	tr	tr	tr
**Terpene alcohol**									
α-Bisabolol	1680	0.11 ± 0.05	0.22 ± 0.09	0.09 ± 0.03	0.13 ± 0.03	0.36 ± 0.04	0.08 ± 0.01	0.08 ± 0.01	-
**Monoterpenes**									
α-Thujene	927	1.02 ± 0.11	0.12 ± 0.02	1.28 ± 0.17	0.86 ± 0.21	1.16 ± 0.11	1.10 ± 0.05	1.10 ± 0.04	3.19 ± 0.24
α-Pinene ^f^	937	19.73 ± 1.72	11.14 ± 1.10	16.65 ± 1.26	22.67 ± 0.45	14.76 ± 0.68	24.20 ± 0.40	20.20 ± 0.69	4.72 ± 0.19
Camphene	949	0.79 ± 0.04	0.46 ± 0.06	0.63 ± 0.02	1.22 ± 0.09	0.70 ± 0.02	0.93 ± 0.02	0.82 ± 0.05	-
sabinene	967	2.75 ± 0.25	3.60 ± 0.15	3.56 ± 0.46	1.46 ± 0.41	2.87 ± 0.40	2.86 ± 0.10	3.07 ± 0.21	7.37 ± 0.62
β-Pinene ^f^	972	44.99 ± 2.09	23.19 ± 2.17	42.28 ± 1.31	46.60 ± 1.89	36.18 ± 0.83	49.51 ± 1.45	43.24 ± 0.19	2.15 ± 0.36
β-Myrcene	983	0.97 ± 0.02	2.26 ± 0.16	0.41 ± 0.02	0.28 ± 0.10	0.64 ± 0.09	0.37 ± 0.01	0.82 ± 0.07	2.40 ± 0.09
δ-3-Carene	998	-	0.85 ± 0.04	-	0.43 ± 0.04	-	-	0.46 ± 0.06	1.92 ± 0.09
α-Terpinene	1007	0.52 ± 0.05	0.05 ± 0.00	0.61 ± 0.07	0.65 ± 0.16	0.63 ± 0.06	0.56 ± 0.03	0.60 ± 0.02	1.64 ± 0.09
Limonene ^f^	1015	7.06 ± 1.17	20.98 ± 1.01	6.45 ± 0.31	4.66 ± 1.96	7.47 ± 1.68	4.29 ± 0.08	9.05 ± 1.05	32.08 ± 2.81
γ-Terpinene	1045	-	0.55 ± 0.09	0.60 ± 0.07	0.38 ± 0.12	-	0.52 ± 0.04	-	1.63 ± 0.09
α-Terpinolene	1081	-	0.39 ± 0.05	-	0.26 ± 0.04	0.46 ± 0.01	0.32 ± 0.01	-	1.14 ± 0.07
**Sesquiterpenes**									
α-Cubebene	1353	-	0.16 ± 0.01	0.16 ± 0.05	0.07 ± 0.02	-	-	-	-
α-Ylangene	1378	tr	tr	tr	0.41 ± 0.23	tr	tr	tr	tr
α-Copaene	1382	0.20 ± 0.03	1.45 ± 0.17	0.24 ± 0.05	0.25 ± 0.14	0.62 ± 0.09	0.32 ± 0.02	0.09 ± 0.01	0.24 ± 0.01
β-Elemene	1392	1.13 ± 0.19	3.26 ± 0.12	3.05 ± 0.35	1.27 ± 0.17	2.60 ± 0.30	0.58 ± 0.06	1.04 ± 0.14	4.38 ± 0.28
β-Bourbonene	1393	-	-	tr	0.08 ± 0.00	-	tr	-	-
α-Cedrene	1401	-	0.16 ± 0.01	-	-	-	-	-	-
*cis*-α-Bergamotene	1414	-	0.49 ± 0.07	-	-	-	-	-	-
α-Gurgujene	1418	-	-	-	0.07 ± 0.00	-	-	-	-
β-Caryophyllene ^f,g,h^	1427	2.68 ± 0.40	8.49 ± 1.16	10.71 ± 1.56	2.74 ± 0.74	10.81 ± 1.03	2.76 ± 0.52	2.61 ± 0.25	12.53 ± 1.51
Aromadendrene	1433	-	-	-	tr	-	-	-	-
α-Bergamotene ^h^	1436	0.77 ± 0.10	1.31 ± 0.25	-	0.55 ± 0.08	0.90 ± 0.01	-	-	0.71 ± 0.04
*cis*-Thujopsene	1437	-	0.15 ± 0.02	-	0.09 ± 0.00	-	-	-	-
β-Gurjunene	1442	-	-	0.37 ± 0.06	-	-	-	-	-
*cis*-β-Farnesene ^h^	1445	-	0.69 ± 0.05	-	0.39 ± 0.11	0.59 ± 0.08	-	-	-
*trans*-β-Farnesene	1452	-	0.20 ± 0.02	-	-	0.18 ± 0.01	-	0.05 ± 0.01	-
α-Caryophyllene ^f,h^	1461	0.87 ± 0.10	1.33 ± 0.18	1.51 ± 0.20	0.49 ± 0.10	2.16 ± 0.31	0.50 ± 0.08	0.59 ± 0.25	1.84 ± 0.33
α-Muurolene	1479	-	0.36 ± 0.13	-	0.16 ± 0.02	-	-	0.10 ± 0.02	-
γ-Muurolene	1479	-	-	-	-	-	-	-	0.17 ± 0.03
Valencene	1486	0.12 ± 0.01	-	-	-	-	-	-	-
Germacrene D	1487	-	-	0.18 ± 0.00	0.11 ± 0.02	-	-	-	-
β-Selinene	1494	0.35 ± 0.06	0.65 ± 0.17	0.56 ± 0.23	2.72 ± 0.33	1.01 ± 0.11	-	0.38 ± 0.17	1.43 ± 0.12
α-Selinene	1501	-	-	-	1.50 ± 0.17	-	-	-	1.17 ± 0.09
β-Bisabolene	1505	0.42 ± 0.10	0.89 ± 0.09	-	tr	1.20 ± 0.16	-	0.34 ± 0.02	-
α-Chamigrene	1514	-	-	-	0.07 ± 0.01	-	-	-	-
α-Amorphene	1518	-	-	-	0.07 ± 0.02	-	-	-	-
δ-Cadinene	1523	-	-	0.15 ± 0.04	0.08 ± 0.02	-	-	-	-
*trans*-γ-Bisabolene	1526	-	0.16 ± 0.00	0.06 ± 0.00	-	-	-	-	-
*trans*-α-Bisabolene	1537	0.18 ± 0.04	0.31 ± 0.05	0.10 ± 0.04	0.43 ± 0.07	-	-	-	-
**Terpene oxide**									
Caryophyllene oxide ^f,g^	1585	-	0.04 ± 0.01	-	-	-	-	-	-
**Hydrocarbons**									
2-Methyl-octane	869	-	-	0.53 ± 0.05	-	-	-	-	-
Undecane	1098	0.51 ± 0.07	0.44 ± 0.11	0.27 ± 0.02	0.12 ± 0.01	0.24 ± 0.02	-	0.25 ± 0.00	-
Dodecane	1197	0.35 ± 0.07	0.19 ± 0.04	-	tr	0.23 ± 0.00	-	-	0.06 ± 0.01
Tridecane	1294	-	0.12 ± 0.06	0.08 ± 0.01	tr	0.09 ± 0.01	-	0.08 ± 0.01	-
**Furan**									
2-Pentylfuran	978	tr	tr	tr	tr	tr	tr	tr	tr

^a^ Identification of components based on the GC/MS library (Wiley 7N). ^b^ Retention indices, using paraffin (C_5_–C_25_) as references. ^c^ Relative percentages from GC-FID, values are means ± standard deviation ( SD) of triplicates. ^d^ Trace. ^e^ Undetectable. ^f^ Published in the literature (Wei et al. [9]). ^g^ Published in the literature (Lu et al. [11]). ^h^ Published in the literature (Deng et al. [10]).

**Table 3 foods-08-00415-t003:** Comparisons of the total volatile compounds in eight varieties of Hsian-tsao extracted using SDE.

Varieties	Concentration (mg/kg) ^a^
Nongshi No. 1	230.71 ± 89.56 ^b^
Taoyuan No. 1	206.90 ± 50.00 ^bc^
Taoyuan No. 2	194.20 ± 50.37 ^bc^
Chiayi Strain	264.61 ± 22.87 ^ab^
TYM1301	329.82 ± 82.32 ^a^
TYM1302	205.43 ± 69.26 ^bc^
TYM1303	124.67 ± 79.29 ^c^
TYM1304	207.50 ± 53.42 ^bc^

^a^ The 100 g samples of Hsian-tsao were extracted using SDE for 2 h, quantification using cyclohexyl acetate as an internal standard. The data correspond to the mean ± SD of triplicates. Values having different superscripts were significantly (*p* < 0.05) different.

**Table 4 foods-08-00415-t004:** Comparisons of volatile compounds from eight varieties of Hsian-tsao extracted using SDE.

Compound ^a^	RI ^b^	Concentration (mg/kg) ^c^							
		Nongshi No. 1	Taoyuan No. 1	Taoyuan No. 2	Chiayi Strain	TYM1301	TYM1302	TYM1303	TYM1304
**Aliphatic alcohols**									
Isobutanol	643	tr ^d^	- ^e^	-	-	-	-	-	-
1-Penten-3-ol	693	tr	tr	tr	tr	tr	tr	tr	tr
1-Pentanol ^f^	739	-	tr	-	-	-	-	-	-
Isoamyl alcohol ^f^	767	0.77 ± 0.89	0.46 ± 0.29	0.16 ± 0.09	0.47 ± 0.29	1.28 ± 0.12	3.10 ± 0.69	1.31 ± 0.82	0.87 ± 0.20
3-Hexanol	791	-	-	0.04 ± 0.01	-	-	-	-	-
(*Z*)-hex-3-en-1-ol	845	1.39 ± 0.53	0.59 ± 0.09	0.77 ± 0.26	0.61 ± 0.03	1.12 ± 0.01	1.29 ± 0.13	0.53 ± 0.22	0.77 ± 0.09
(*E*)-2-hexen-1-ol	855	0.02 ± 0.00	tr	0.02 ± 0.03	0.01 ± 0.02	-	0.20 ± 0.11	0.05 ± 0.04	-
Hexanol ^f^	856	0.34 ± 0.26	0.22 ± 0.05	0.36 ± 0.08	0.41 ± 0.54	0.19 ± 0.03	0.45 ± 0.52	0.23 ± 0.03	0.17 ± 0.01
1-Octen-3-ol ^f^	962	0.44 ± 0.25	0.32 ± 0.00	0.31 ± 0.07	0.50 ± 0.12	1.14 ± 0.15	0.69 ± 0.02	0.23 ± 0.15	1.05 ± 0.06
3-Octanol ^f^	979	0.07 ± 0.03	tr	0.06 ± 0.03	0.07 ± 0.01	-	0.12 ± 0.10	0.14 ± 0.06	0.16 ± 0.02
**Aliphatic aldehydes**									
Pentanal ^f^	700	tr	tr	tr	tr	tr	tr	tr	tr
Hexanal ^f^	777	0.69 ± 0.59	tr	0.03 ± 0.03	0.14 ± 0.04	0.22 ± 0.14	5.07 ± 4.73	0.37 ± 0.40	0.07 ± 0.03
(*E*)-2-hexenal ^f^	832	2.63 ± 1.36	0.48 ± 0.08	0.19 ± 0.06	0.22 ± 0.02	0.86 ± 0.01	0.69 ± 0.11	0.54 ± 0.27	0.96 ± 0.08
(*Z*)-4-heptenal	877	0.02 ± 0.01	tr	0.04 ± 0.01	0.04 ± 0.00	0.01 ± 0.01	0.16 ± 0.06	tr	0.11 ± 0.08
(*E*,*E*)-2,4-hexadienal	879	0.05 ± 0.02	tr	tr	tr	0.05 ± 0.02	tr	tr	0.04 ± 0.01
(*E*)-2-heptenal	931	tr	tr	tr	0.03 ± 0.00	0.01 ± 0.01	0.05 ± 0.00	tr	tr
(*E*,*E*)-2,4-heptadienal ^f^	979	tr	-	-	tr	0.14 ± 0.02	tr	-	-
(*E*)-2-octenal ^f^	1024	0.39 ± 0.09	0.45 ± 0.08	0.43 ± 0.08	0.56 ± 0.07	0.37 ± 0.04	0.72 ± 0.04	0.42 ± 0.21	0.50 ± 0.09
Nonanal ^f^	1078	3.99 ± 1.18	0.11 ± 0.00	0.18 ± 0.00	0.16 ± 0.02	1.07 ± 0.17	0.93 ± 0.03	tr	2.20 ± 0.27
(*E*)-2-nonenal ^f^	1130	0.34 ± 0.22	0.23 ± 0.01	0.06 ± 0.11	0.65 ± 0.08	0.51 ± 0.03	0.82 ± 0.05	0.45 ± 0.17	0.69 ± 0.12
Safranal	1175	0.80 ± 0.22	tr	0.73 ± 0.20	-	0.54 ± 0.03	-	-	0.76 ± 0.17
Decanal ^f^	1181	-	0.34 ± 0.30	0.32 ± 0.04	0.21 ± 0.03	0.12 ± 0.03	0.34 ± 0.01	0.23 ± 0.07	0.16 ± 0.02
(*E*)-2-decenal	1241	0.40 ± 0.21	0.19 ± 0.03	0.32 ± 0.07	-	0.12 ± 0.05	0.64 ± 0.01	0.33 ± 0.05	0.17 ± 0.08
(*E*,*E*)-2,4-decadienal ^f^	1285	0.20 ± 0.16	0.56 ± 0.58	1.17 ± 0.49	1.07 ± 0.14	0.42 ± 0.08	1.48 ± 0.04	0.85 ± 0.52	0.89 ± 0.25
**Aliphatic ketones**									
3-Hexen-2-one	820	-	-	0.07 ± 0.04	-	-	-	-	-
2-Heptanone ^f^	872	0.18 ± 0.04	tr	0.10 ± 0.03	0.26 ± 0.02	0.17 ± 0.08	0.36 ± 0.05	0.22 ± 0.13	0.17 ± 0.09
1-Octen-3-one ^f^	956	4.79 ± 1.63	1.97 ± 0.03	1.89 ± 0.76	2.19 ± 0.31	6.47 ± 0.43	3.49 ± 0.14	2.66 ± 0.93	8.29 ± 0.94
3-Octanone	965	2.43 ± 2.56	tr	0.25 ± 0.05	2.97 ± 1.25	tr	5.51 ± 1.15	0.45 ± 0.31	tr
3,5-Octadien-2-one	1066	0.16 ± 0.02	0.07 ± 0.00	0.10 ± 0.00	0.09 ± 0.01	0.08 ± 0.03	0.30 ± 0.12	0.22 ± 0.13	0.11 ± 0.04
2-Nonanone	1070	0.28 ± 0.08	-	-	0.37 ± 0.05	-	0.51 ± 0.05	0.29 ± 0.13	-
6-Methyl-3,5-heptadien-2-one ^f^	1074	1.26 ± 0.17	1.08 ± 0.51	1.04 ± 0.39	2.07 ± 0.30	0.73 ± 0.02	1.30 ± 0.07	1.63 ± 0.43	0.93 ± 0.14
3-Nonen-2-one	1110	0.18 ± 0.06	0.13 ± 0.01	0.15 ± 0.02	-	-	0.26 ± 0.04	0.17 ± 0.07	-
β-Damascenone ^f,g^	1365	1.17 ± 0.45	0.70 ± 0.02	1.50 ± 0.33	0.97 ± 0.09	0.63 ± 0.55	1.87 ± 0.02	1.03 ± 0.37	0.51 ± 0.16
**Aliphatic ester**									
Ethyl acetate	631	-	-	tr	tr	tr	tr	tr	tr
**Aromatic alcohols**									
Benzyl alcohol ^f^	999	0.18 ± 0.06	tr	0.10 ± 0.02	0.18 ± 0.01	0.12 ± 0.02	0.57 ± 0.05	0.35 ± 0.23	0.11 ± 0.06
1-Octanol	1048	0.18 ± 0.05	tr	0.10 ± 0.01	0.13 ± 0.01	-	0.36 ± 0.12	-	-
Eugenol ^g^	1328	2.16 ± 0.96	1.04 ± 0.28	0.62 ± 0.44	2.17 ± 0.25	1.27 ± 0.29	0.73 ± 0.02	0.35 ± 0.18	1.00 ± 0.35
Methyleugenol	1369	0.23 ± 0.19	0.15 ± 0.06	2.43 ± 0.49	0.44 ± 0.07	-	-	-	-
**Aromatic aldehydes**									
Benzaldehyde ^f^	933	tr	tr	tr	tr	tr	tr	tr	tr
Benzeneacetaldehyde	1002	tr	0.11 ± 0.01	tr	tr	0.20 ± 0.01	0.80 ± 0.33	tr	0.28 ± 0.04
**Aromatic ketone**									
Acetophenone	1033	0.46 ± 0.07	0.29 ± 0.03	0.38 ± 0.03	0.31 ± 0.04	0.03 ± 0.02	0.62 ± 0.05	0.26 ± 0.12	0.05 ± 0.02
**Aromatic esters**									
Hexyl acetate	991	tr	-	-	-	-	-	-	-
Methyl salicylate	1170	1.29 ± 0.49	0.82 ± 0.11	1.78 ± 0.25	-	0.56 ± 0.08	-	1.70 ± 0.48	0.42 ± 0.09
**Aromatic hydrocarbons**									
*p*-Cymene	1004	0.30 ± 0.08	0.29 ± 0.01	0.27 ± 0.03	0.31 ± 0.07	-	-	0.34 ± 0.25	0.24 ± 0.05
**Aromatic hydrocarbons**									
β-Methylnaphthalene	1275	0.30 ± 0.20	-	-	0.39 ± 0.05	-	-	-	0.25 ± 0.14
α-Ionene ^g^	1397	-	0.45 ± 0.11	-	-	-	-	-	-
Cuparene	1514	-	-	-	-	-	-	2.14 ± 2.60	10.00 ± 5.81
**Terpene alcohols**									
Linalool ^f^	1082	0.20 ± 0.07	-	0.21 ± 0.00	0.18 ± 0.06	0.15 ± 0.03	0.34 ± 0.04	0.21 ± 0.11	0.13 ± 0.05
Borneol ^g^	1149	0.63 ± 0.24	0.26 ± 0.12	0.35 ± 0.09	-	-	-	0.35 ± 0.29	-
Menthol	1157	-	-	-	-	-	-	0.19 ± 0.07	-
4-Terpineol ^f^	1163	0.48 ± 0.23	0.17 ± 0.01	0.26 ± 0.03	0.58 ± 0.09	0.08 ± 0.03	0.83 ± 0.17	0.28 ± 0.17	0.13 ± 0.04
α-Terpineol ^f^	1173	0.27 ± 0.17	0.27 ± 0.16	tr	1.58 ± 0.28	0.17 ± 0.04	1.45 ± 0.22	tr	0.17 ± 0.05
Myrtenol	1184	0.18 ± 0.13	-	0.11 ± 0.09	0.19 ± 0.03	tr	1.14 ± 0.04	0.64 ± 0.27	-
Gossonorol	1617	0.99 ± 0.77	2.03 ± 0.11	-	-	1.51 ± 0.46	-	1.81 ± 0.83	1.28 ± 0.51
α-Bisabolol	1680	58.69 ± 18.60	72.25 ± 12.19	4.23 ± 0.16	116.29 ± 3.79	144.20 ± 32.40	2.78 ± 0.10	42.90 ± 15.95	0.67 ± 0.29
**Terpene aldehydes**									
β-Cyclocitral	1198	0.83 ± 0.42	-	1.41 ± 0.13	0.84 ± 0.15	0.91 ± 0.07	2.60 ± 0.06	0.65 ± 0.32	1.11 ± 0.22
β-Citral ^f^	1217	-	-	-	-	-	-	-	0.07 ± 0.06
**Terpene ketones**									
Camphor	1128	-	0.43 ± 0.07	0.65 ± 0.26	-	-	-	-	-
β-Ionene^g^	1469	4.84 ± 3.53	-	3.10 ± 0.55	3.29 ± 0.28	2.46 ± 0.46	8.28 ± 0.48	1.30 ± 1.06	4.09 ± 1.17
**Monoterpenes**									
α-Thujene	927	0.63 ± 0.83	0.47 ± 0.24	0.34 ± 0.32	0.80 ± 0.65	4.45 ± 0.54	1.28 ± 0.52	1.00 ± 1.46	0.62 ± 0.09
α-Pinene ^f^	937	0.02 ± 0.02	tr	tr	0.03 ± 0.00	0.05 ± 0.05	0.04 ± 0.00	tr	0.03 ± 0.01
Camphene	949	0.87 ± 0.32	0.96 ± 0.02	0.67 ± 0.21	1.18 ± 0.04	4.16 ± 0.30	2.48 ± 0.17	1.18 ± 0.74	7.17 ± 0.46
Sabinene	967	-	1.56 ± 0.52	1.83 ± 1.43	-	12.57 ± 1.41	-	3.38 ± 4.04	1.41 ± 0.23
β-Pinene ^f^	972	2.39 ± 0.66	0.73 ± 0.26	1.12 ± 0.26	0.97 ± 0.20	0.59 ± 0.05	1.66 ± 0.10	0.95 ± 0.53	-
β-Myrcene	983	-	tr	-	-	-	-	-	0.06 ± 0.02
α-Phellandrene	998	-	0.13 ± 0.00	0.21 ± 0.01	-	-	0.22 ± 0.20	-	-
Limonene ^f^	1015	0.70 ± 0.23	1.43 ± 0.13	0.63 ± 0.12	0.90 ± 0.12	1.41 ± 1.14	0.57 ± 0.03	0.59 ± 0.35	2.43 ± 2.05
γ-Terpinene	1045	0.28 ± 0.07	0.23 ± 0.01	0.27 ± 0.11	0.29 ± 0.03	0.23 ± 0.03	0.37 ± 0.00	-	0.16 ± 0.02
α-Terpinolene	1081	-	0.11 ± 0.01	-	-	-	-	-	-
**Sesquiterpenes**									
α-Cubebene	1353	0.28 ± 0.22	0.26 ± 0.09	0.51 ± 0.13	0.48 ± 0.01	-	1.10 ± 0.27	0.30 ± 0.22	-
α-Ylangene	1378	0.82 ± 0.39	1.80 ± 0.13	0.80 ± 0.07	1.17 ± 0.11	1.04 ± 0.60	2.66 ± 0.14	0.31 ± 0.20	0.99 ± 0.31
α-Copaene	1382	0.47 ± 0.33	0.69 ± 0.20	1.60 ± 0.20	1.04 ± 0.09	-	1.65 ± 0.00	0.75 ± 0.41	1.76 ± 1.77
β-Elemene	1392	8.07 ± 3.08	7.73 ± 0.54	16.89 ± 4.16	-	11.94 ± 2.44	-	4.48 ± 2.02	35.25 ± 8.75
β-Bourbonene	1393	-	-	-	6.78 ± 0.57	-	6.40 ± 0.07	-	-
β-Caryophyllene ^f,g,h^	1427	24.11 ± 9.73	20.57 ± 3.12	55.71 ± 16.29	18.99 ± 2.29	53.59 ± 10.44	20.68 ± 18.88	4.13 ± 1.55	52.80 ± 10.04
α-Bergamotene ^h^	1436	7.64 ± 1.99	3.51 ± 0.16	-	5.36 ± 0.30	5.90 ± 1.07	-	-	5.91 ± 1.86
*cis*-Thujopsene	1437	-	-	-	-	-	-	1.32 ± 1.01	-
β-Gurjunene	1442	tr	-	-	-	-	-	-	5.90 ± 1.83
*cis*-β-Farneseneh	1445	2.57 ± 0.89	1.86 ± 0.51	-	-	0.79 ± 0.04	-	-	-
α-Caryophyllene ^f,h^	1461	6.73 ± 3.56	5.50 ± 0.77	3.85 ± 3.46	5.68 ± 0.12	11.83 ± 2.62	8.57 ± 2.49	2.59 ± 1.53	12.51 ± 2.53
γ-Muurolene	1479	-	2.55 ± 0.22	-	-	-	-	-	-
Germacrene D	1487	4.77 ± 3.40	5.12 ± 0.77	7.63 ± 1.31	6.03 ± 0.26	5.73 ± 1.74	-	-	6.40 ± 1.65
β-Selinene	1494	11.48 ± 2.20	5.92 ± 2.25	5.20 ± 1.26	22.45 ± 2.48	8.32 ± 6.00	7.31 ± 0.89	5.42 ± 2.70	tr
α-Selinene	1501	5.68 ± 0.88	3.96 ± 2.85	10.22 ± 1.00	tr	12.66 ± 8.28	8.14 ± 1.48	5.91 ± 2.72	14.45 ± 3.13
β-Bisabolene	1505	10.74 ± 2.11	8.08 ± 3.25	-	23.72 ± 0.53	8.76 ± 3.09	6.39 ± 2.50	-	-
δ-Guaiene	1510	-	-	-	-	-	-	-	3.93 ± 0.92
α-Chamigrene	1514	2.14 ± 1.82	-	-	4.60 ± 0.02	-	-	2.37 ± 3.11	-
γ-Cadinene	1517	11.17 ± 3.59	3.98 ± 3.80	7.72 ± 4.82	6.38 ± 1.42	-	13.13 ± 1.01	-	-
δ-Cadinene	1523	-	8.54 ± 9.04	4.23 ± 1.65	6.14 ± 0.39	-	8.52 ± 3.31	4.40 ± 6.68	2.15 ± 1.45
*trans*-γ-Bisabolene	1526	5.81 ± 7.90	3.43 ± 3.46	-	4.47 ± 4.18	0.56 ± 0.17	-	-	2.68 ± 1.83
*trans*-α-Bisabolene	1537	3.38 ± 2.03	3.49 ± 0.37	-	3.97 ± 0.22	6.90 ± 1.60	-	2.99 ± 3.22	-
**Terpene oxide**									
Caryophyllene oxide ^f,g^	1585	24.16 ± 5.73	27.53 ± 1.85	32.15 ± 3.37	2.13 ± 0.21	8.67 ± 4.24	49.59± 20.74	15.07 ± 9.85	9.51 ± 1.81
**Hydrocarbons**									
Heptane	721	tr	-	-	-	-	-	-	-
Octane	805	-	-	-	0.03 ± 0.00	-	0.05 ± 0.05	-	-
Nonane	898	-	tr	0.06 ± 0.01	0.07 ± 0.01	0.04 ± 0.00	0.11 ± 0.02	-	0.02 ± 0.01
Undecane	1098	0.21 ± 0.06	0.21 ± 0.07	0.13 ± 0.03	-	0.06 ± 0.03	-	-	-
Dodecane	1197	-	-	-	-	0.24 ± 0.02	-	-	-
Tridecane	1294	-	-	-	-	0.13 ± 0.07	-	-	-
Pentadecane	1498	-	-	-	-	-	12.41 ± 6.36	-	-
**Straight-chain acids**									
Nonanoic acid	1253	tr	-	0.22 ± 0.04	-	-	-	0.22 ± 0.17	-
Decanoic acid	1344	-	-	5.32 ± 4.16	-	-	1.25 ± 0.06	tr	-
Dodecanoic acid	1542	tr	-	tr	-	-	-	1.22 ± 1.14	-
**Furans**									
2-Ethylfuran	712	tr	tr	-	-	tr	tr	tr	tr
Furfural	795	0.02 ± 0.00	tr	tr	-	tr	-	0.04 ± 0.04	tr
2-Pentylfuran	978	0.15 ± 0.04	-	0.02 ± 0.02	-	0.90 ± 0.17	-	-	1.28 ± 0.46
**Methoxy-phenolic compounds**									
2-Methoxy-phenol	1058	-	0.09 ± 0.00	0.14 ± 0.02	-	0.04 ± 0.02	-	-	-
2-Methoxy-4-vinylphenol	1279	tr	tr	0.19 ± 0.02	-	0.18 ± 0.05	-	-	0.35 ± 0.16
**Nitrogen-containing compound**									
Indole	1260	0.22 ± 0.16	0.23 ± 0.01	0.25 ± 0.03	-	0.17 ± 0.11	-	0.23 ± 0.16	0.15 ± 0.10

^a^ Identification of components based on the GC/MS library (Wiley 7N); ^b^ Retention indices, using paraffin (C_5_–C_25_) as references. ^c^ Relative percentages from GC-FID, values are means ± SD of triplicates; ^d^ Trace; ^e^ Undetectable. ^f^ Published in the literature (Wei et al. [9]); ^g^ Published in the literature (Lu et al. [11]); ^h^ Published in the literature (Deng et al. [10]).

**Table 5 foods-08-00415-t005:** Percentages of extracted chemical groups of Hsian-tsao analyzed using HS-SPME.

	Nongshi No. 1	Taoyuan No.1	Taoyuan No. 2	Chiayi Strain	TYM1301	TYM1302	TYM1303	TYM1304
Aliphatic alcohol	0.29	0.69	1.16	0.17	2.25	1.39	1.61	3.01
Aliphatic aldehydes	3.63	2.91	0.29	0.39	1.19	1.15	1.81	1.74
Aliphatic ketones	tr	0.14	0.12	tr	0.29	tr	0.24	0.47
Aliphatic ester	0.32	-	-	-	-	-	-	-
Aromatic aldehyde	tr	-	-	tr	-	-	-	-
Aromatic alcohol	-	-	-	0.08	-	-	-	-
Aromatic hydrocarbon	-	0.68	-	tr	tr	tr	tr	tr
Terpene alcohol	0.11	0.22	0.09	0.13	0.36	0.08	0.08	-
Monoterpenes	77.83	63.59	72.47	79.47	64.87	84.63	79.36	58.24
Sesquiterpenes	6.72	20.06	16.93	11.28	20.5	4.16	5.2	22.47
Terpene oxide	-	0.04	-	-	-	-	-	-
Hydrocarbons	0.86	0.75	0.88	0.12	0.56	-	0.33	0.06
Furan	tr	tr	tr	tr	tr	tr	tr	tr

All the definitions of the symbols used in Table 2 mean values were also used in Table 5.

**Table 6 foods-08-00415-t006:** Concentrations of chemical groups of overall extracted Hsian-tsao analyzed using SDE.

	Nongshi No. 1	Taoyuan No. 1	Taoyuan No. 2	Chiayi Strain	TYM1301	TYM1302	TYM1303	TYM1304
Aliphatic alcohols	3.03	1.59	1.72	2.07	3.73	5.85	2.49	3.02
Aliphatic aldehydes	9.51	2.36	3.47	3.08	4.44	10.90	3.19	6.55
Aliphatic ketones	10.45	3.95	5.10	8.92	8.08	13.60	6.67	10.01
Aliphatic ester	-	-	tr	tr	tr	tr	tr	tr
Aromatic alcohols	2.75	1.19	3.25	2.92	1.39	1.66	0.7	1.11
Aromatic aldehydes	tr	0.11	tr	tr	0.2	0.8	tr	0.28
Aromatic ketone	0.46	0.29	0.38	0.31	0.03	0.62	0.26	0.05
Aromatic esters	1.29	0.82	1.78	-	0.56	-	1.7	0.42
Aromatic hydrocarbons	0.60	0.74	0.27	0.7	-	-	2.48	10.49
Terpene alcohols	61.44	74.98	5.16	118.82	146.11	6.54	46.38	2.38
Terpene aldehydes	0.83	-	1.41	0.84	0.91	2.6	0.65	1.18
Terpene ketones	4.84	0.43	3.75	3.29	2.46	8.28	1.3	4.09
Monoterpenes	4.89	5.62	5.07	4.17	23.46	6.62	7.1	11.88
Sesquiterpenes	105.86	86.99	124.36	117.26	128.02	84.55	34.97	144.73
Terpene oxide	24.16	27.53	32.15	2.13	8.67	49.59	15.07	9.51
Hydrocarbons	0.21	0.21	0.19	0.10	0.47	12.57	0	0.02
Straight-chain acids	tr	-	5.54	-	-	1.25	1.44	-
Furans	0.17	tr	0.02	-	0.9	-	0.04	1.28
Methoxy-phenolic compounds	tr	0.09	0.33	-	0.22	-	-	0.35
Nitrogen-containing compound	0.22	-	0.25	-	0.17	-	0.23	0.15

Notes: All the definitions of the symbols used in Table 3 mean values were also used in Table 4.

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
