# Peer review of "Analysis of Volatile Constituents in Platostoma palustre (Blume) Using Headspace Solid-Phase Microextraction and Simultaneous Distillation-Extraction"

_foods, 2019, doi:10.3390/foods8090415_

Round 1

Reviewer 1 Report

This is a very interesting study on the determination of the volatile component of eight varieties of Hsian-tsao.  GC-MS was the analysis method with solid phase microextraction (SPME) and simultaneous distillation extraction (SDE) as the sample preparation methods.  The sample preparation methods were complementary.  This should be of interest to the readers of Food.  Please improve your manuscript based on the comments below.

Figure 3 described peak heights. This should be in terms of amounts found (compound) per gram of sample. This will improve the quality of the presentation. The recoveries of the targeted analytes (at least the important ones) using the sample preparation methods should be conducted. Without the recovery values, it is difficult to interpret the data, or what was actual amount found per gram of sample.  The recovery studies for HS-SPME and SDE will be beneficial for other scientist who would like to choose which method is appropriate for their studies. The data in Table 4 should include the recovery of the extraction procedure. Please provide a discussion on why HS-SPME extracted more monoterpenes; while SDE extracted more sesquiterpenes, terpene alcohols, and terpene oxides. Also, a reasoning why SDE was able to detect more components.

Author Response

Thank you for your useful comments and suggestions on the language and structure of our manuscript. The manuscript has been revised as suggested. Revised portions have been marked with red letters, and detailed corrections are listed below point by point:

1) Figure 3 described peak heights. This should be in terms of amounts found (compound) per gram of sample. This will improve the quality of the presentation. The recoveries of the targeted analytes (at least the important ones) using the sample preparation methods should be conducted. Without the recovery values, it is difficult to interpret the data, or what was actual amount found per gram of sample. The recovery studies for HS-SPME and SDE will be beneficial for other scientist who would like to choose which method is appropriate for their studies.

Response: Thank you for your useful comments and suggestions. Figure 3 described total peak areas not peak heights. The total peak areas we used may be not the best for this study, but it still is very good for comparison purposes. Reineccius (2006) indicated that no method will accurately reflect the aroma compounds actually present in a food or their proportions. IF one wants an overall view of aroma profile, then one will likely combine several methods of aroma isolation, each isolation tenchnique providing a part of the overall profile. SPME affords a certain view of the volatile composition of food. The method provides an isolate has the compounds to measure and it is adequately reproducible, the method is quite attractive. There are no solvents for contamination; it is simple, automated, moderately sensitive and rapid.

Reference:
Reineccius, G. 2006 Flavor chemistry and technology, 2nd ed.; CRC Press, Taylor & Francis Group FL, USA, 2006; 41–52.
2) The data in Table 4 should include the recovery of the extraction procedure.
Response: We have revised and showed in Table 4.
2
3) Please provide a discussion on why HS-SPME extracted more monoterpenes; while SDE extracted more sesquiterpenes, terpene alcohols, and terpene oxides. Also, a reasoning why SDE was able to detect more components.
Response: We have revised.

Reviewer 2 Report

GENERAL

1a
Consider the usage of the current name of this plant -> Platostoma palustre (Blume) A. J. Paton.
Check http://www.worldfloraonline.org/taxon/wfo-0000274811

1b
Check the present knowledge/bibliography concerning at last Mesona chinensis Benth., Mesona palustris Blume, and Mesona parviflora (Benth.) Briq. that are botanically defined to be synonymous to previously mentioned Latin name. Some refs were just cited.

Regardless, discuss the topic with a botanist and try to explain a problem of nomenclatural complexity briefly in the Intro part. Current botanical references can be cited.

2
GENERAL NOVELTY is missing or hidden. You have a plant that was previously analyzed by GC-MS (-> Refs). You have a well-known method. And, you have some varieties that are newly analyzed. Think about something eye-catching in the Abstract and the Conclusion. Maybe any practical application of your results? Avoid too many numbers.

MAJOR & CONTEXT

l.18-19
'It is admired by many consumers especially in summer, because of its aroma and taste' -> Consider developing the Discussion: If the plant is popular - it is probably propagated by consumers for their own needs. If so - it is very likely to find many varieties differing with aroma strength and properties. Then, your work is only a very starting point. Also, there is something to do in the future.

l.27-20
'Compared with SDE, HS-SPME sampling extracted a significantly higher amount of monoterpenes and had poorer detection of less volatile compounds, such as sesquiterpenes, terpene alcohols, and terpene oxide'

Is it a new discovery? -> Better, think what really new has been demonstrated by your work and describe this in Abstract instead.

l.34
To enrich the value of this work, use the proper name of this plant not only in pinyin, phonetic-English (Hsian-tsao, Liangfen Cao) but also in Chinese or Vietnamese characters. I'm not a language expert and cannot judge this, but as the Author, supervise the final version of the manuscript to be sure that the proper characters were adequately used by the Editor.

l.64
It is hard to discuss 'aromatic notes' if the notes were not 'scientifically' investigated. I mean - there is no methodology considering this topic in the manuscript. Thus, it looks like 'so-called aromatic notes' or 'everyone knows what I'm writing about'. -> Try to use other words or supply the work.

ch.2.1.
What about vouchers? -> If no botanical vouchers were made, conserve at least samples of herbs that can be sent to other investigators (You should also declare this at the end of the manuscript, about line 402, see template).

l.86-89
Am I right that the sample was subjected to HS-SPME as dry sample? -> State.

l.114, 121

Is DB-1 made with fused-silica? -> Think about the source of mistake and correct.

l.137
'extracted more total volatile components than the other' How was it quantified? There is no info on the internal standard in SPME procedure (2.2.1+2.2.2.). -> Supply the paragraph in the Experimental section. Optionally, cite Silva & Co. [22] there.

l.141
Is the volatiles profile in cocoa similar to volatiles in Hsian-tsao samples? Can be compared? -> Comment, at least.

ch.3.1.1., 3.1.2.
Beside total peak areas, were there any specific differences among classes of cmpds? Think about something like 'fatty-derived peaks were smallest on fiber A while biggest on D and oxygenated sesquiterpenes area order was D>E>C>A>B' and so on. If applicable.

Do the same with the time parameter (hydrocarbon cmpds at Fig.2. seems to be on the same level at 30-40-50 min when total looks like increasing at 40 min).

ch.3.2.
Double digits are not necessary when reporting approximate ranges. E.g., consider '(43.2-49.5%)' or even '(43-50%)' instead of '(43.24-49.51%)' and similar.

Similarly in chapter 3.3
If you look at '58.69-144.20 mg/kg', you can express it briefly in Results as '59-144 mg/kg' without unnecessary precision. Precise results can be found in Tables.

l.218
'α-Bisabolol displayed antifungal, anti-inflammatory and analgesic activities'. The following statement has no sense in the context of this paragraph. You have about 145mg/a-bisabolol/kg of herb that is equal to example 0.7mg/a-bisabolol/5g serving of culinary herb seasoning. What was the effective dose and way of application of a-bisabolol in cited reference? -> Avoid.

l.236-238
Similar to the above. Think about the comparison of activity (at literature concentration and way of application) vs. reported concentration of caryophyllene oxide. Removing or rewriting this will not diminish the value of your work.

l.219
How to define 'aroma' in 'aroma concentration'? We can measure 'aroma intensity' by panel experts, but I would not advise stating 'aroma'='total concentration of VOCs'. Think about another expression.

+l.234
O.K., now I see that you are still writing on the bisabolol. Regardless, change the doubtful word and remove paragraph fist-line-spacing here.

EDITORIAL

l.21, 31, 51
Check the position of hyphen in 'simultaneous-distillation extraction (SDE)' -> 'simultaneous distillation-extraction (SDE)'

31, 169
Check the position of hyphen (if any) in 'headspace-solid phase microextraction (SPME)' -> 'headspace solid-phase microextraction (SPME)'

l.75-76
'After harvest, postharvest treatments were performed in air drying shade' -> Try to simplify the sentence.

l.97
-> Name the fiber invariably
l.85 (DVB/CAR/PDMS) vs. l.97 (PDMS)

l.98
'Each sample [...]' -> 'Each sample, homogenized as described above in 2.2.1. [...]'?

l.111-112
-> Solvent evaporation conditions (temp., time, column type, and length) should be precised.

l.114
-> Define the producer of DB-1 column. If also Agilent, just mention.

l.191-192
'Nongshi No. 1 contained the highest content about hexanal (3.63%)' -> Try to use another word instead 'about'.

Fig.3. and its caption
'Strain' is used mainly in microbiology. Do you mean 'varieties'? -> Change the words. Check the manuscript looking for the same.

Fig.3. and its caption
'Values having different superscripts are significantly (p < 0.05) different' -> No visible superscripts at all. Supply or change.

l.198
'content' -> Do you mean 'number'? The content=total area is highest in Nongshi No.1 and lowest in TYM1304 as presented in the table.

BTW
Can you compare the total yield of VOCs to the literature results from SDE or to yield of essential oil by steam- or hydrodistillation?

Table 2 and other long tables
Ask technical Editor for better reproduction of table 2 (see -> table breaking and captions).

Table 2 and the following tables
There is info in a footer that a) identification is based on Wiley library -> add 'RI', 'fragmentation pattern' or both. -> Add percentages info -> from GC-FID or GC-MS(?)

Table 3 caption
'Percentages of chemical groups of oval Hsian-tsao' - What does mean 'oval Hsian-tsao'? Is it a new term?

l.245
remove repetition 'content...'

ch.3.4
Consider: '12 [-> add 'mainly ...'] were identified by HS-SPME but not detected by SDE, and 64 [-> add 'mainly ...'] were identified by SDE but not detected by HS-SPME.

ch.3.4
It is hard to find in this chapter which results are yours and which are from literature to comparison. -> Rewrite clearly. Some examples below:

l.249-251
'Additionally, only α-terpineol was found by the SDE method' Where? In [37]? You have reported 64 cmpds in SDE over SPME a few lines before. Rewrite clearly. 'Additionally, they have found that...'

l.256-257
'[...] that SDE used high temperature and long extraction time and large quantities of volatile components were lost [...]' -> '[...] that when SDE was conducted in high temperature [-> REPORT their temperature in C degrees here] and for a long extraction time [-> REPORT their time of SDE here], large quantities of volatile components were lost [...]'

BTW. Did they use for SDE the same solvent as you? Comment.

l.266
'[...] is complementary to traditional methods for the determination of the volatile compounds in Chinese herbs'

Your subject is Lamiaceae member that is typical to yield volatiles and SDE is generally a good strategy to analyze it. Emphasizing ' in Chinese herbs' look forced here. Consider rewriting.

Final 1
Check all work for lacking ('DB-1(60 m ×0 .25 mm') or unnecessary (e.g., '36.18 - 42.28%') spaces between characters.

Final 2
Check all the work for the homogeneity of British/American (-> usage of s/z)

REFS

[8]
There is no such article in Food Sci. Technol. 2014, 322 39, (5), 190-192.
However, it is in 'Shipin Keji'. https://www.worldcat.org/title/xiandai-shipin-keji-modern-food-science-and-technology/ -> Check and correct to an international abbreviation for this journal. -> Check other references to remove possible similar mistakes.

Author Response

Dear reviewers’
Thank you for your useful comments and suggestions on the language and structure of our manuscript. The manuscript has been revised as suggested. Revised portions have been marked with red letters, and detailed corrections are listed below point by point:

1a. Consider the usage of the current name of this plant -> Platostoma palustre (Blume) A. J. Paton. Check http://www.worldfloraonline.org/taxon/wfo-0000274811

Response: Thank you for your useful comments and suggestions. We have checked and revised.

1b. Check the present knowledge/bibliography concerning at last Mesona chinensis Benth., Mesona palustris Blume, and Mesona parviflora (Benth.) Briq. that are botanically defined to be synonymous to previously mentioned Latin name. Some refs were just cited. Regardless, discuss the topic with a botanist and try to explain a problem of nomenclatural complexity briefly in the Intro part. Current botanical references can be cited.

Response: Thank you very much for your suggestions. We have cited.

2. GENERAL NOVELTY is missing or hidden. You have a plant that was previously analyzed by GC-MS (-> Refs). You have a well-known method. And, you have some varieties that are newly analyzed. Think about something eye-catching in the Abstract and the Conclusion. Maybe any practical application of your results? Avoid too many numbers.

Response: Thank you very much for your suggestions. We have revised.

l.18-19: 'It is admired by many consumers especially in summer, because of its aroma and taste' -> Consider developing the Discussion: If the plant is popular - it is probably propagated by consumers for their own needs. If so - it is very likely to find many varieties differing with aroma strength and properties. Then, your work is only a very
2
starting point. Also, there is something to do in the future.

Response: Thank you for your comments. In the future, we will also study on effect of storage and concentration on the processing adaptability of Hsian-tsao and gelling property among different Hsian-tsao varieties.

l.27-20: Compared with SDE, HS-SPME sampling extracted a significantly higher amount of monoterpenes and had poorer detection of less volatile compounds, such as sesquiterpenes, terpene alcohols, and terpene oxide. Is it a new discovery? -> Better, think what really new has been demonstrated by your work and describe this in Abstract instead.

Response: Thank you for your useful comments. We have revised and showed in abstract.

l.64: It is hard to discuss 'aromatic notes' if the notes were not 'scientifically' investigated. I mean - there is no methodology considering this topic in the manuscript. Thus, it looks like 'so-called aromatic notes' or 'everyone knows what I'm writing about'. -> Try to use other words or supply the work.

Response: Thank you for your comments. We have revised.

ch.2.1 : What about vouchers? -> If no botanical vouchers were made, conserve at least samples of herbs that can be sent to other investigators (You should also declare this at the end of the manuscript, about line 402, see template).

Response: Thank you for your comments. We have declared this at the end of the manuscript.

l.86-89: Am I right that the sample was subjected to HS-SPME as dry sample? -> State.

Response: Yes, in our country, traditional Hsian-tsao processing always uses dried Hsian-tsao.

l.114, 121: Is DB-1 made with fused-silica? -> Think about the source of mistake and correct.

Response: We have revised.

l.137: 'extracted more total volatile components than the other' How was it quantified? There is no info on the internal standard in SPME procedure (2.2.1+2.2.2.). -> Supply the paragraph in the Experimental section. Optionally, cite Silva & Co. [22] there.

Response: Thank you for your comments. In the 2.2.1 Experimental section, we have

3
cited Yeh et al. [20].
l.141: Is the volatiles profile in cocoa similar to volatiles in Hsian-tsao samples? Can be compared? -> Comment, at least.

Response: Cocoa and Hsian-tsao are food ingredients, the SPME method available for the analysis of volatiles of food samples (Kataoka et al., 2000)

Reference:
Kataoka, H.; Lord, H.L.; Pawliszyn, J. Applications of solid-phase microextraction in food analysis. J. Chromatogr. A 2000, 880, (1-2) 35-62.
ch.3.1.1., 3.1.2: Beside total peak areas, were there any specific differences among classes of cmpds? Think about something like 'fatty-derived peaks were smallest on fiber A while biggest on D and oxygenated sesquiterpenes area order was D>E>C>A>B' and so on. If applicable.

Response: Thank you for your suggestions. The total peak areas we used may be not the best for this study, but it still is very good for comparison purposes.

Do the same with the time parameter (hydrocarbon cmpds at Fig.2. seems to be on the same level at 30-40-50 min when total looks like increasing at 40 min).

Response: Total peak areas, not only hydrocarbons but also alcohol, ketones, and aldehydes, etc.

ch.3.2: Double digits are not necessary when reporting approximate ranges. E.g., consider '(43.2-49.5%)' or even '(43-50%)' instead of '(43.24-49.51%)' and similar.

Response: We have revised.

Similarly in chapter 3.3: If you look at '58.69-144.20 mg/kg', you can express it briefly in Results as '59-144 mg/kg' without unnecessary precision. Precise results can be found in Tables.

Response: We have revised.

l.218: 'α-Bisabolol displayed antifungal, anti-inflammatory and analgesic activities'. The following statement has no sense in the context of this paragraph. You have about
4
145mg/a-bisabolol/kg of herb that is equal to example 0.7mg/a-bisabolol/5g serving of culinary herb seasoning. What was the effective dose and way of application of a-bisabolol in cited reference? -> Avoid

Response: We have revised.

l.236-238: Similar to the above. Think about the comparison of activity (at literature concentration and way of application) vs. reported concentration of caryophyllene oxide. Removing or rewriting this will not diminish the value of your work.

Response: We have removed.

l.219: How to define 'aroma' in 'aroma concentration'? We can measure 'aroma intensity' by panel experts, but I would not advise stating 'aroma'='total concentration of VOCs'. Think about another expression.

Response: We have revised.

l.234: O.K., now I see that you are still writing on the bisabolol. Regardless, change the doubtful word and remove paragraph fist-line-spacing here.

Response: We have revised.

EDITORIAL
l.21, 31, 51: Check the position of hyphen in 'simultaneous-distillation extraction (SDE)' -> 'simultaneous distillation-extraction (SDE)’

Response: We have checked and revised.

31, 169: Check the position of hyphen (if any) in 'headspace-solid phase microextraction (SPME)' -> 'headspace solid-phase microextraction (SPME)'

Response: We have checked and revised.

l.75-76: 'After harvest, postharvest treatments were performed in air drying shade' -> Try to simplify the sentence.

Ans: We have revised.

l.97: Name the fiber invariably, l.85 (DVB/CAR/PDMS) vs. l.97 (PDMS)

Response: We have revised.
5
l.98: 'Each sample [...]' -> 'Each sample, homogenized as described above in 2.2.1. [...]'

Response: We have revised.

l.111-112: Solvent evaporation conditions (temp., time, column type, and length) should be precised.

Response: Thank you for your useful comments and suggestions. We have revised.

l.114: Define the producer of DB-1 column. If also Agilent, just mention.

Response: We have revised.

l.191-192: 'Nongshi No. 1 contained the highest content about hexanal (3.63%)' -> Try to use another word instead 'about'.

Response: We have revised.

Fig.3. and its caption: 'Strain' is used mainly in microbiology. Do you mean 'varieties'? -> Change the words. Check the manuscript looking for the same.

Response: We have revised and show in Fig. 3.

Fig.3. and its caption: 'Values having different superscripts are significantly (p < 0.05) different' -> No visible superscripts at all. Supply or change.

Response: We have deleted 'Values having different superscripts are significantly (p < 0.05) different'.

l.198: 'content' -> Do you mean 'number'? The content=total area is highest in Nongshi No.1 and lowest in TYM1304 as presented in the table.

Response: We mean 'concentration' showed in Table 4.

BTW: Can you compare the total yield of VOCs to the literature results from SDE or to yield of essential oil by steam- or hydrodistillation?

Response: In Table 4, do the same with the study parameter and added an internal standard, internal standard was used to obtain the weight concentration of volatile compound in the sample. So, we can compare the total yield of VOCs to the literature results from SDE.
6
Table 2 and other long tables: Ask technical Editor for better reproduction of table 2 (see -> table breaking and captions).

Response: We have revised.

Table 2 and the following tables: There is info in a footer that a identification is based on Wiley library -> add 'RI', 'fragmentation pattern' or both. -> Add percentages info -> from GC-FID or GC-MS(?)

Response: Thank you for your review. We have revised.

Table 3 caption: 'Percentages of chemical groups of oval Hsian-tsao' - What does mean 'oval Hsian-tsao'? Is it a new term?

Response: We have revised.

l.245: remove repetition 'content...'

Response: We have removed.

ch.3.4: Consider: '12 [-> add 'mainly ...'] were identified by HS-SPME but not detected by SDE, and 64 [-> add 'mainly ...'] were identified by SDE but not detected by HS-SPME.

Response: We have revised.

ch.3.4: It is hard to find in this chapter which results are yours and which are from literature to comparison. -> Rewrite clearly. Some examples below:
l.249-251: 'Additionally, only α-terpineol was found by the SDE method' Where? In [37]? You have reported 64 cmpds in SDE over SPME a few lines before. Rewrite clearly. 'Additionally, we have found that...'

Response: We have revised.

l.256-257: '[...] that SDE used high temperature and long extraction time and large quantities of volatile components were lost [...]' -> '[...] that when SDE was conducted in high temperature [-> REPORT their temperature in C degrees here] and for a long extraction time [-> REPORT their time of SDE here], large quantities of volatile components were lost [...]'
BTW. Did they use for SDE the same solvent as you? Comment.
7
Response: In [37] they use for SDE solvent was prepared by n-pentane/dichloromethane, we use n-pentane/ diethyl ether, although the composition is different, but there are similar solvent polarity characteristics.

l .266: '[...] is complementary to traditional methods for the determination of the volatile compounds in Chinese herbs'. Your subject is Lamiaceae member that is typical to yield volatiles and SDE is generally a good strategy to analyze it. Emphasizing ' in Chinese herbs' look forced here. Consider rewriting.

Response: We now have rewritten.

Final 1: Check all work for lacking ('DB-1(60 m ×0 .25 mm') or unnecessary (e.g., '36.18 - 42.28%') spaces between characters.
Response: We now have checked and revised.
Final 2: Check all the work for the homogeneity of British/American (-> usage of s/z)

Response: We now have checked

REFS [8]: There is no such article in Food Sci. Technol. 2014, 322 39, (5), 190-192. However, it is in 'Shipin Keji'. https://www.worldcat.org/title/xiandai-shipin-keji-modern-food-science-and-technology/ -> Check and correct to an international abbreviation for this journal. -> Check other references to remove possible similar mistake.
Response: REFS [Wei et al., 2014] is from China journal (ISSN: 1005-9989). The English name of the journal is Food Science and Technology. There is no an international abbreviation for this journal. Although it is not an international journal, the volatile aroma components of Platostoma palustre (Blume) is rarely reported in international journals.

Round 2

Reviewer 2 Report

PREVIOUS REVISION COMMENTS

>>l.219: How to define 'aroma' in 'aroma concentration'? We can measure 'aroma intensity' by panel experts, but I would not advise stating 'aroma'='total concentration of VOCs'. Think about another expression.

>Response: We have revised.

! still in l.29

>>Fig.3. and its caption: 'Strain' is used mainly in microbiology. Do you mean 'varieties'? -> Change the words. Check THE MANUSCRIPT looking for the same.

>Response: We have revised and show in Fig. 3.

Fig.3 caption

! Avoid 'strain' in this context -> use 'variety'.

Tab.4 heading

! Avoid 'strain' in this context -> use 'variety'.

! Look elsewhere!

>>REFS [8]: There is no such article in Food Sci. Technol. 2014, 322 39, (5), 190-192. However, it is in 'Shipin Keji'. https://www.worldcat.org/title/xiandai-shipin-keji-modern-food-science-and-technology/ -> Check and correct to an international abbreviation for this journal. -> Check other references to remove possible similar mistake.

>Response: REFS [Wei et al., 2014] is from China journal (ISSN: 1005-9989). The English name of the journal is Food Science and Technology. There is no an international abbreviation for this journal. Although it is not an international journal, the volatile aroma components of Platostoma palustre (Blume) is rarely reported in international journals.

! OK, then I would advise to use both Chinese (in transcription) and English full name without abbreviation. Else, discuss this with Technical Editor.

MAIN NOTES

l.47

'effective' -> 'supposed to be effective' or 'recommended by herbalists'

because [4] and [5] report the data from in vitro experiments and not from experiments considering living organisms suffering from 'heat-shock, hypertension, diabetes, liver diseases and muscle and joint pains'.

paragraph, starting at l.49

As there is no wider botanical intro previously, consider finishing this paragraph with 'Mesona chinensis is also considered as a synonym of P. palustris by botanists [e.g., 1]'.

paragraph, starting at l.121

At the end of paragraph clear for what purpose GC-FID was used. E.g., 'for RI comparison' or 'for quantitation of peak areas' or both.

EDITORIAL NOTES

l.22-27

The sense of paragraph is clear. The language not.

-> Consult with language specialist/native speaker.

l.40

'call' -> 'called'

l.56

-> Consider changing '3-allyl-6-methoxyphenol' to 'chavibetol' (common, popular name of this volatile cmpd)

200-203

Very clearly supplied. But it is still not stated what method (GC or GCMS) was used to calculate areas.

-> Simply, do it in main GC/GCMS chapter once as I suggested above and avoid here and elsewhere (btw., thanks for clearing this at the long table as c)

Tab.3

-> Change 'a)' and 'b)' in table body into subs/supers.

l.261-262

-> The new sentence needs language care.

l.257 and 259

-> Avoid 'extraction' word; it is just in abbreviations SDE and SPME.

Tab.4

-> Change 'a)', 'b)' and 'c)' in table body into subs/supers.

Good luck with further experiments!

Author Response

Dear reviewers’
Thank you for your useful comments and suggestions on the language and structure of our manuscript. The manuscript has been revised as suggested. Revised portions have been marked with red letters, and detailed corrections are listed below point by point:
For Reviewer:
>>l. 219: How to define 'aroma' in 'aroma concentration'? We can measure 'aroma intensity' by panel experts, but I would not advise stating 'aroma'='total concentration of VOCs'. Think about another expression. still in l.29
Response: We have revised.
Fig.3. and its caption: 'Strain' is used mainly in microbiology. Do you mean 'varieties'? -> Change the words. Check THE MANUSCRIPT looking for the same.
>Response: We have revised and show in Fig. 3.
Fig.3 caption
! Avoid 'strain' in this context -> use 'variety'.
Response: We have revised.
Tab.4 heading
! Avoid 'strain' in this context -> use 'variety'.
! Look elsewhere!
Response: We have revised.
>>REFS [8]: There is no such article in Food Sci. Technol. 2014, 322 39, (5), 190-192. However, it is in 'Shipin Keji'. https://www.worldcat.org/title/xiandai-shipin-keji-modern-food-science-and-technology/ -> Check and correct to an international abbreviation for this journal. -> Check other references to remove possible similar mistake.
>Response: REFS [Wei et al., 2014] is from China journal (ISSN: 1005-9989). The English name of the journal is Food Science and Technology. There is no an
2
international abbreviation for this journal. Although it is not an international journal, the volatile aroma components of Platostoma palustre (Blume) is rarely reported in international journals.
! OK, then I would advise to use both Chinese (in transcription) and English full name without abbreviation. Else, discuss this with Technical Editor.
Response: Thank you for your suggestions. We have revised and showed in references [9].
MAIN NOTES
l.47
'effective' -> 'supposed to be effective' or 'recommended by herbalists'
because [4] and [5] report the data from in vitro experiments and not from experiments considering living organisms suffering from 'heat-shock, hypertension, diabetes, liver diseases and muscle and joint pains'.
Response: We have revised.
paragraph, starting at l.49 As there is no wider botanical intro previously, consider finishing this paragraph with 'Mesona chinensis is also considered as a synonym of P. palustris by botanists [e.g., 1]'.
Response: We have revised.
paragraph, starting at l.121
At the end of paragraph clear for what purpose GC-FID was used. E.g., 'for RI comparison' or 'for quantitation of peak areas' or both.
Response: The purpose GC-FID was used both for RI comparison and quantitation of peak areas.
EDITORIAL NOTES
l.22-27
The sense of paragraph is clear. The language not.
-> Consult with language specialist/native speaker.
Response: We have revised.
3
l.40
'call' -> 'called'
Response: We have revised.
l.56-> Consider changing '3-allyl-6-methoxyphenol' to 'chavibetol' (common, popular name of this volatile cmpd)
Response: We have revised.
200-203
Very clearly supplied. But it is still not stated what method (GC or GCMS) was used to calculate areas.
-> Simply, do it in main GC/GCMS chapter once as I suggested above and avoid here and elsewhere (btw., thanks for clearing this at the long table as c)
Response: We have revised.
Tab.3
-> Change 'a)' and 'b)' in table body into subs/supers.
Response: We have revised.
l.261-262
-> The new sentence needs language care.
Response: We have revised.
l.257 and 259
-> Avoid 'extraction' word; it is just in abbreviations SDE and SPME.
Response: We have revised.
Tab.4
-> Change 'a)', 'b)' and 'c)' in table body into subs/supers.
Response: We have revised.
Good luck with further experiments!
Response: Thank you for your review and suggestion.
